# Single-cell network biology characterizes cell type gene regulation for drug repurposing and phenotype prediction in Alzheimer's disease

**Chirag Gupta**[1,2], **Jielin Xu**[3], **Ting Jin**[1,2], **Saniya Khullar**[1,2], **Xiaoyu Liu**[4], **Sayali Alatkar**[1,5], **Feixiong Cheng**[3,6,7], **Daifeng Wang**[1,2,5] *

1 Waisman Center, University of Wisconsin-Madison, Madison, Wisconsin, United States of America,
2 Department of Biostatistics and Medical Informatics, University of Wisconsin-Madison, Madison, Wisconsin, United States of America, 3 Genomic Medicine Institute, Lerner Research Institute, Cleveland Clinic, Cleveland, Ohio, United States of America, 4 Department of Statistics, University of Wisconsin-Madison, Madison, Wisconsin, United States of America, 5 Department of Computer Sciences, University of Wisconsin-Madison, Madison, Wisconsin, United States of America, 6 Department of Molecular Medicine, Cleveland Clinic Lerner College of Medicine, Case Western Reserve University, Cleveland, Ohio, United States of America, 7 Case Comprehensive Cancer Center, Case Western Reserve University School of Medicine, Cleveland, Ohio, United States of America

* daifeng.wang@wisc.edu

**Data Availability Statement:** All our results are provided in Supplementary Datasets. The cell type GRNs predicted in this study are deposited to a

## Abstract

Dysregulation of gene expression in Alzheimer's disease (AD) remains elusive, especially at the cell type level. Gene regulatory network, a key molecular mechanism linking transcription factors (TFs) and regulatory elements to govern gene expression, can change across cell types in the human brain and thus serve as a model for studying gene dysregulation in AD. However, AD-induced regulatory changes across brain cell types remains uncharted. To address this, we integrated single-cell multi-omics datasets to predict the gene regulatory networks of four major cell types, excitatory and inhibitory neurons, microglia and oligodendrocytes, in control and AD brains. Importantly, we analyzed and compared the structural and topological features of networks across cell types and examined changes in AD. Our analysis shows that hub TFs are largely common across cell types and AD-related changes are relatively more prominent in some cell types (e.g., microglia). The regulatory logics of enriched network motifs (e.g., feed-forward loops) further uncover cell type-specific TF-TF cooperativities in gene regulation. The cell type networks are also highly modular and several network modules with cell-type-specific expression changes in AD pathology are enriched with AD-risk genes. The further disease-module-drug association analysis suggests cell-type candidate drugs and their potential target genes. Finally, our network-based machine learning analysis systematically prioritized cell type risk genes likely involved in AD. Our strategy is validated using an independent dataset which showed that top ranked genes can predict clinical phenotypes (e.g., cognitive impairment) of AD with reasonable accuracy. Overall, this single-cell network biology analysis provides a comprehensive map linking genes, regulatory networks, cell types and drug targets and reveals cell-type gene dysregulation in AD.

public repository available at https://zenodo.org/record/5829585. The codes for our analyses are available at https://github.com/daifengwanglab/scNET.

**Funding:** This work was supported by National Institutes of Health (NIH) grants R01AG067025, R21CA237955, R03NS123969, R21NS127432 and U01MH116492 to D.W., R01AG066707, R01AG076448, U01AG073323, 3R01AG066707-02S1 and 1R56AG074001-01 to F.C., National Science Foundation Career Award 2144475 to D. W., and the start-up funding for D.W. from the Office of the Vice Chancellor for Research and Graduate Education at the University of Wisconsin–Madison. This study was supported in part by a core grant to the Waisman Center from the National Institute of Child Health and Human Development (NIH P50HD105353). The funders had no role in study design, data collection and analysis, decision to publish, or preparation of the manuscript.

**Competing interests:** The authors have declared that no competing interests exist.

## Author summary

Alzheimer's Disease (AD) is the leading cause of dementia. It affects parts of the brain that control language, behavior, and memory. The human brain is comprised of tens of billions of cells, such as neuronal cells that transmit information via electrical and chemical signals, and glial cells that maintain the brain's immune system. Researchers have found that AD causes changes in the expression of genes within the brain cells. Gene expression is a tightly regulated process involving interconnected networks of multiple genes. Understanding how these gene networks change in AD is critical to identifying genetic biomarkers and potential drug targets. Using genomic data of post-mortem brains diagnosed with AD and healthy individuals, we identified gene networks that play a crucial role in regulating biological processes within neuronal and glial cells. We utilized these gene networks to make predictions on existing FDA approved drugs that could potentially be repurposed for AD. Furthermore, we used a machine learning strategy to identify novel genes that are more likely to be involved in AD pathology. The systems-level approach lends itself to analysis of single-cell genomics data of other human diseases.

## Introduction

Alzheimer's Disease (AD) is a brain disorder that progresses into memory loss, a decline in cognitive skills, and ultimately dementia. The mechanistic causes of AD are not yet fully understood, especially at the cell type level, although the abnormal accumulation of neuronal tangles and amyloid plaques in the AD brain have become potential hallmarks of the disease. The genetic factors that possibly lie upstream of various AD phenotypes have now been extensively studied from next generation sequencing data, such as genome-wide gene expression changes. A variety of computational analyses have been applied to those data for understanding abnormal gene expression and regulation in AD. However, most studies have been performed on bulk tissue data and missed cell type specific signals. The neurovascular unit as a whole could drive AD progression [1], and recent studies have verified that molecular changes in AD are highly cell type-specific [2]. Thus, it is imperative to investigate the contribution of individual cell types in the brain to the progression of AD along with clinical phenotypes. Emerging single-cell RNA-seq (scRNA-seq) enables such an analysis, as it captures the transcriptomic landscape of individual cells, offering a rich source of data for the analysis of dysregulated molecular systems within individual cells.

Several studies have highlighted strong links between molecular connectivity and human diseases, suggesting that disease risk genes often work together as a coherent biological network. Thus, it is critical to study broken functional relationships between genes, rather than individual genes, to better understand the molecular mechanisms associated with the disease. Network biology offers a powerful computational framework that transcends individual gene investigation that uses univariate methods, such as differential expression analysis. For example, gene regulatory networks (GRNs) provide information about regulatory interactions between regulators, e.g., transcription factors (TFs), and their potential target genes. Such GRN models can be used to derive novel biological hypotheses about dysregulated disease pathways. With scRNA-seq data quickly accumulating in open repositories, single-cell network biology is now leading a shift from the traditional bulk RNA-seq mediated analyses [3–7]. Although GRNs in AD have been previously explored using expression data from bulk

tissues [8–13], cell type level GRN in AD remains under-investigated, especially via network biology approaches.

Network biology has been successfully applied to prioritize novel disease genes. The basic idea is to identify regulatory genes that have more influence over the network by virtue of their network position. Naturally, a more prominent position in the network is occupied by hubs or genes with a relatively larger number of connections and those that facilitate signalling between distant genes in the network. Hubs play a central role in modulating the expression of many genes and thus biological processes and pathways. Network biologists have adopted various classical metrics from graph theory to identify hubs in GRNs. Network-based indicators of gene importance have also been useful for the analysis of disease at the cell type level. For example, Iacono et al. analyzed healthy and diabetic pancreatic cell networks and found that genes involved in type-2 diabetes differ in their centralities scores [14]. In addition, single-cell network analyses have revealed genes that rewire with exposure to differentiation cues [15] and cancer-causing perturbations [16]. In the context of brain diseases, single-cell gene networks have indicated a potential cell type preference of neuropsychiatric and neurodegenerative disorders [17] and neurodevelopmental disorders (NDDs) [18]. The authors in the later study estimated coexpression between sets of known NDD-risk genes and demonstrated that most gene sets have higher coexpression in neural progenitor cells, suggesting a convergent role of these cell types in NDDs [18].

The structure of gene regulatory networks can also help understand coordinated gene regulation. Recurring sub-graphs, called network motifs, are patterns that appear in real networks more often than random networks. Network motifs are the building blocks of biological networks. Therefore, comparing network motifs in, for example, control and disease states of the transcriptome can unravel how the disease affects the structural design of the GRN. For example, the feed-forward loop (FFL) is a three-node motif particularly interesting for analysis directed networks [19,20]. FFLs comprise a master regulator, which regulates an intermediate TF, and both TFs directly regulate the expression of a common target gene. This information can help gauge changes in 'regulatory pressure' on downstream TFs through coordinated activities between upstream TFs. Network motifs can also be helpful in applying digital computing ideas such as logic gates in synthetic biology [21,22].

Network-medicine is an upcoming field to solve the problem of drug repurposing by finding new uses of existing drugs by linking them to drug targets which are also implicated in human diseases [23–26]. Network-medicine approaches have been applied to repurpose drug candidates for cancers [27], tuberculosis [28], and, more recently, respiratory illnesses like COVID-19 [29,30]. We have also previously developed network-medicine strategies for AD. For example, we recently proposed an endophenotype network-based drug repurposing framework for AD [31]. Our approach uses disease-associated modules (modules enriched with disease genes) and network proximity analysis for in silico drug repurposing. Using this approach, we discovered sildenafil as a new candidate drug for AD, tested it using insurance record data, and validated it using iPSCs from patients with AD [31]. Our study shows that quantifying the network distance between AD modules and drug targets in the human interactome can significantly improve in silico drug discovery.

In this study, we analyzed and compared GRN characteristics of the human brain cell types and examined regulatory changes that occur in AD (Fig 1). We integrated available single-nucleus gene expression (snRNA-seq), single-cell chromatin interaction, and open-chromatin (ATAC-seq) data to predict cell type GRNs. Specifically, we linked TFs to TGs if the putative DNA binding motif of a given TF is located in the open and interacting promoter or enhancer region of the TG, and if the TF has a certain degree of coexpression with the TG in a given cell type. Thus, TFs are linked to TGs via enhancers and promoters in two neuronal and two glial

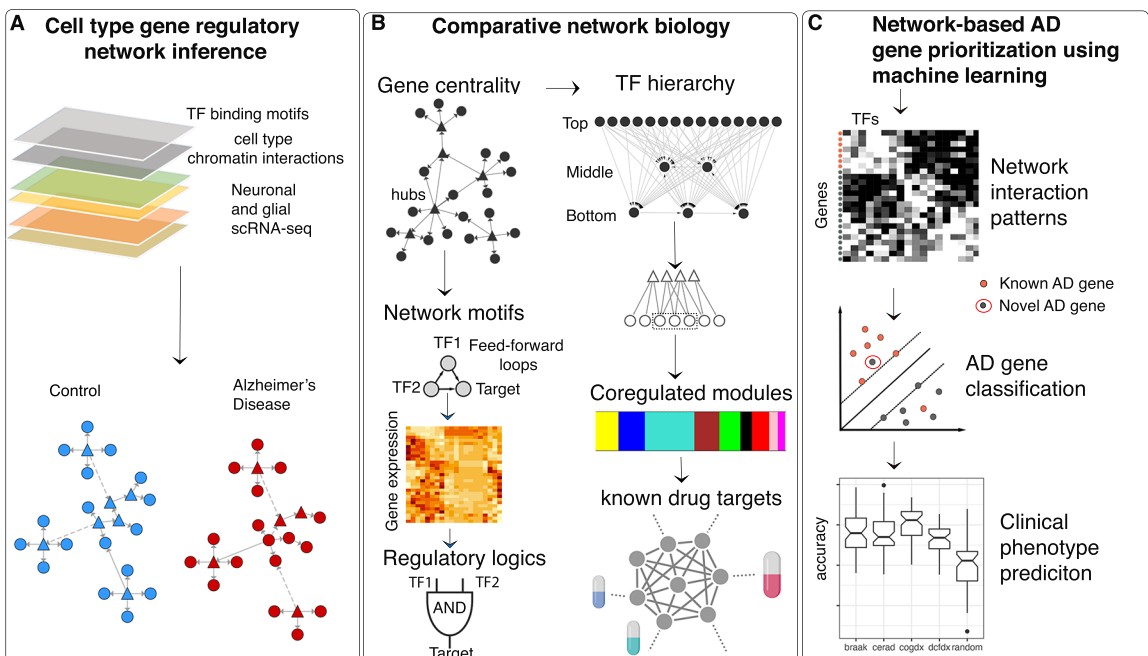

**Fig 1. An integrative network-biology framework for analyzing cell type gene regulatory mechanisms in Alzheimer's disease (AD).** (**A**) Predicting neuronal and glial cell type GRNs from multi-omics data in AD and control by integrating snRNA-seq with chromatin interaction data and TF binding site information. (**B**) Analyses of cell type GRN characteristics include identification of hub genes, regulatory hierarchy, network motifs, regulatory logics, modules of co-regulated genes, and drug-module associations. (**C**) Machine learning based prioritization of novel AD genes using network interaction patterns and prediction of clinical phenotypes using machine learning.

cell types in control and AD conditions (Fig 1A). Then, we compared topological features such as gene centrality, structural features such as network motifs and logic gates, and the modular organization of each cell type GRN. We adopted a network-proximity strategy to demonstrate the utility of network modules for identification of likely drug candidates for AD (Fig 1B). Finally, we utilized known AD genes from the published literature in a machine-learning analysis to generate a ranking of genes according to their potential association with AD. We validated these rankings on an independent RNA-seq dataset (ROSMAP) (Fig 1C).

## Results

We applied our analytic framework to single-cell multi-omics data for four cell types, excitatory and inhibitory neurons, microglia, and oligodendrocytes from human brains diagnosed with Alzheimer's disease (AD) and healthy controls [2]. All detailed descriptions on datasets and data processing are available in Methods and Materials. The dataset consists of single-nucleus RNA-sequencing (snRNA-seq) samples from the prefrontal cortex of 24 individuals diagnosed with AD and 24 age-matched controls with no AD pathology. In addition, we also obtained cell-type chromatin interactions [32], cell-type open chromatin regions [33], and human transcription factor binding site information [34]. We predicted GRNs for four major each cell types for which all three data modalities were available. First, we predicted all possible interactions between enhancers and promoters for each cell type using the chromatin interaction data. Then, we inferred the transcription factor binding sites (TFBS) based on consensus binding site sequences in the interacting enhancers and promoters. We connected TFs to TGs via enhancers only if the enhancers are highly accessible in the ATAC-seq data. Subsequently,

we retained TF-TG links that have high gene expression relationships in snRNA-seq data. Overall, we created separate control and AD GRNs for each cell type. Additionally, we also obtained single-cell transcriptomic data of healthy cells from an independent study [35] to check the reproducibility of our results.

## Hubs of the brain cell type gene regulatory networks

We linked transcription factors (TFs), non-coding regulatory elements, and target genes to predict cell type GRNs in control and AD. On average, ~17% of all edges in the cell type GRNs link TFs to TG via promoters, ~68% via enhancers, and ~15% via both, enhancers, and promoters (Fig A1 in S1 File). GRNs typically have a nonuniform distribution of links (edges) [36], system biologists are often interested in identifying 'hub' nodes (genes) for practical applications [37]. Hubs represent highly connected genes that have a greater influence over the network. Such highly connected hub genes often play a crucial role in modulating gene expression changes, and thus disease-associated pathways. Given that gene expression phenotype in AD is highly cell type specific [2], we asked if distinct or similar sets of genes act as hubs across cell type networks.

We used three standard centrality metrics to quantify the influence of a given TF over each cell type's control and AD network. The out-degree centrality calculates the number of targets for each TF, in-degree indicates how strongly a TF is under the regulatory influence of other TFs, and the betweenness centrality of a TF is a function of its out-degree and in-degree and estimates its ability to act as a communication channel between upstream regulators and downstream pathway genes. We observed that TFs with the highest out-degrees (top 10% of the sorted list) are largely common (173 TFs) across all cell types (Fig 2A and S1 Data). However, distinct TFs represent betweenness centralities of different cell type GRNs. Moreover, the overlap of such TFs is relatively higher between neuronal than glial cell types (Fig 2B). We noted that TFs that have the greatest regulatory influence of other TFs (high in-coming degrees) also vary across cell types (Fig A2 in S1 File). The overlap of such TFs is also larger between neuronal than glial cell types (Fig A2 in S1 File; see discussion). However, it is important to note that the chromatin interaction data source we used does not resolve between the neuronal cell types. Therefore, the observed variation between excitatory and inhibitory neuron GRNs are purely based on differences in gene expression patterns, unlike other cell types for which changes could also be attributed to distinct E-P interactions.

Although the normalized gene centralities (including non-TF genes) between control and AD GRNs across all cell types are largely correlated, there is a clear differential in the in-degrees, with the most prominent scatter in microglia (Fig 2C). For example, the DNER, RHOU, and SLC1A2 genes have fewer regulators in AD compared to control, whereas RUNDC3A and NPTX1 are regulated by more TFs in AD than in control (Fig 2D). DNER activates the NOTCH1 pathway which is linked to AD [38,39]. SLC1A2 mediates cellular uptake of glutamate, and loss of function of glutamate transporters has been linked to AD [40]. NPTX1 is a member of the pentraxin family, known to modulate synaptic transmission in normal conditions [41]. Also, we noted that microglia GRNs have the largest number of distinct high betweenness TFs. Therefore, we were interested to investigate if the subnetworks around these central TFs have identical or disjoint node- and edge-sets. Visualizing the network neighborhoods of the 23 high betweenness microglia TFs, we observed a considerable difference between the control and AD networks (Fig 2E). This indicated presence of a disease-driven regulatory apparatus governed by the same TFs.

Overall, genes with the largest in-degree changes are significantly enriched in (1% FDR based on hypergeometric tests) in gene ontology (GO) biological process (BP) terms related to

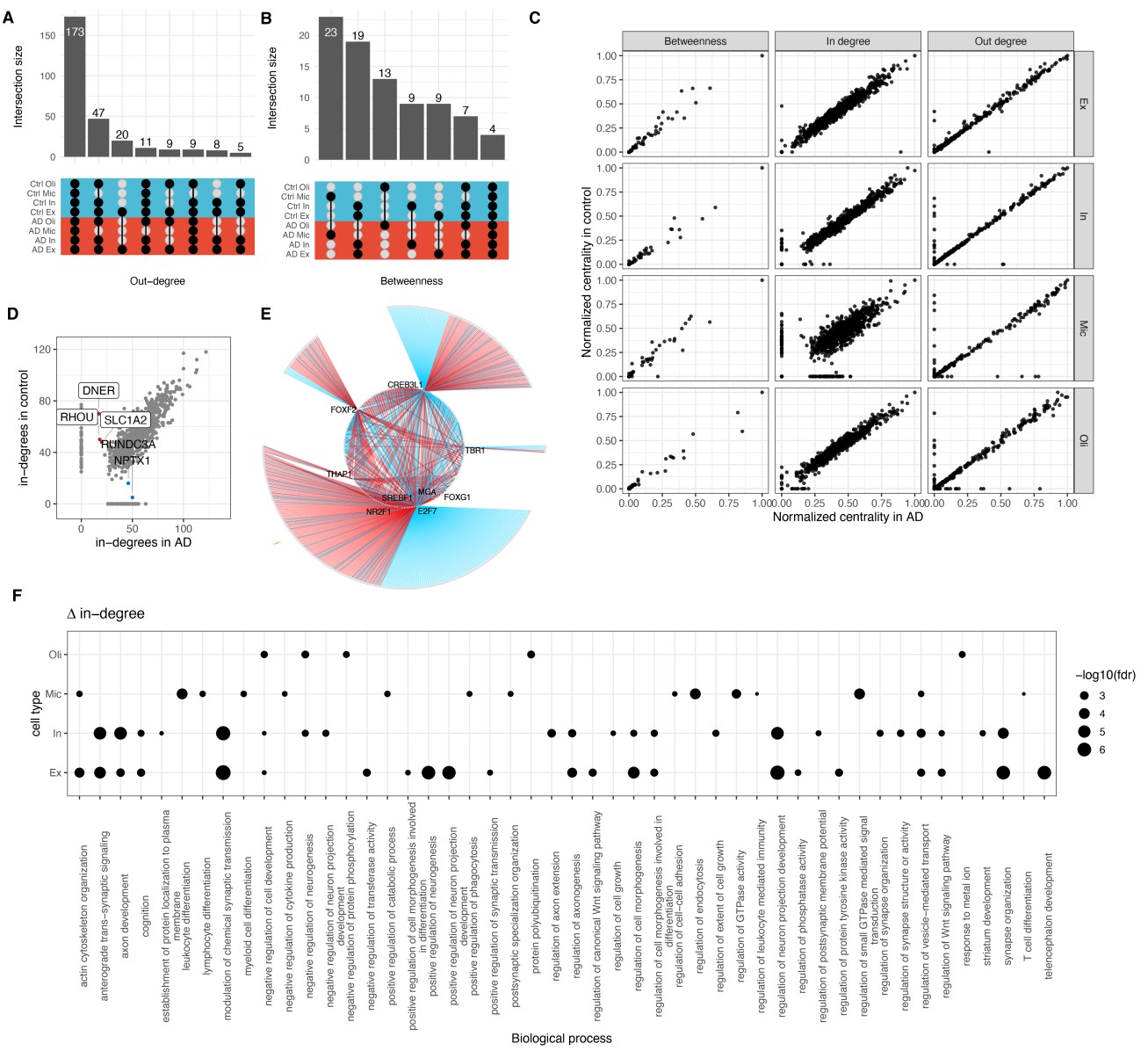

**Fig 2. Centrality analysis reveal hub gene changes of cell-type gene regulatory networks in AD.** (**A**) An upset plot showing overlaps between the top 10% genes with largest out-degree and (**B**) betweenness centralities. The filled dots in the center matrix indicate the comparison between the respective sets (along the x-axis), and the bars on the top show size of the intersection. Blue and red rows indicate control and AD, respectively. (**C**) Scatter plots showing normalized gene centralities distribution and (**D**) the distribution of in-degrees in microglial AD and control networks. Genes with large changes in in-degree between AD and control are labelled. (**E**) Visualization of the subnetwork of 9 TFs with high betweenness centrality in microglia. Grey circles around the periphery of the network indicate target genes. Symbols of the nine central TFs are shown and the rest hidden for clarity. Blue and red edges indicate interaction in the control and AD networks, respectively. (**F**) A dot plot showing enrichment of gene ontology biological processes (y-axis) among genes with the most extensive changes in the in-degree centrality across all cell types (x-axis). The dot size is set according to the FDR-corrected *p* values, as shown in the key.

the immune system such as 'neutrophil-mediated immunity' and 'leukocyte migration' in microglia, development-related processes such as 'autonomic nervous system development' and 'anterior-posterior pattern specification' in oligodendrocytes, and synapse related processes such as 'synapse organization', 'regulation of synaptic transmission', and 'modulation of chemical synaptic transmission' in neuronal cell types (Fig 2F).

To check if our centrality analysis is reproducible in an independent dataset, we obtained data from an earlier study that analyzed gene expression in healthy brain cell types [35]. We applied our GRN inference pipeline and centrality analysis to microglia and oligodendrocytes in this secondary dataset and found that the top central genes (top 20% across all centralities) show high and statistically significant overlaps (permutation-test $P$-value $< 0.001$) with the main dataset (Fig A3 in S1 File). Because the secondary dataset has samples only from healthy brain cells, we could only compare the overlap with our control networks. Nevertheless, the high overlaps indicated that the top central genes we identified are indeed independent of the dataset specificity. Furthermore, the correlation in centrality scores of TFs is high between the full dataset and a reduced dataset comprising of only 50% of the samples from the original snRNA-seq dataset (Fig A4 in S1 File).

## The regulatory hierarchy of brain cell type gene regulatory networks

Given that GRNs are typically hierarchical in structure [42–44], we asked if AD induces changes to the regulatory hierarchy of cell type GRNs. We wanted to identify TFs that act as master regulators, and other TFs that function downstream of the master regulators. Master regulators are defined as TFs at the top of the network hierarchy with no regulatory influence from other TFs [45].

To classify TFs at different levels of regulatory hierarchy, we used the standard hierarchy height (*hh*) metric [44]. According to the *hh* metric, TFs at the top levels of the hierarchy exhibit many outgoing edges but no incoming edges (master regulators not regulated by other TF), TFs at the middle levels exhibit both incoming and outgoing edges (regulators and regulated by other TFs) and TFs at the bottom levels exhibit no outgoing edges to other TFs (highly regulated by other TFs). We found the distribution of normalized *hh* to be trimodal across all GRNs and significantly different from random networks (estimated using the KS test of 1000 random networks) (Fig 3A and S2 Data), indicating that the brain cell type GRNs are indeed hierarchical. We also noted that the *hh* of TFs is not significantly different in control and AD networks (Fig B1 in S1 File). We found 85 (27.6%) master regulators common across all cell type AD GRNs, with the most unique master regulators in excitatory neurons (37 TFs; 12%) (Fig B2 in S1 File). Some common master regulators include known AD genes, such as CREB1, ESR1, HSF1, PPARG, NFE2L2, SPI1, TCF3, TCF7L2, TP53, CLOCK, and GLIS3. However, we found very few TFs in the middle-level (5 TFs in microglia, 3 in excitatory neurons, 1 in inhibitory neurons, and none in oligodendrocytes; Fig B3 in S1 File; see Discussion).

We were interested in knowing if the readjustment of the targets of TFs at various levels of the regulatory hierarchy contributes to AD. We estimated the rewiring score of TFs based on the overlap between their predicted targets in control and AD networks (see Methods). Within the four cell types we analyzed, we found that TFs are least rewired in inhibitory neurons and most in microglia (Fig 3B). Interestingly, neuronal TFs at all levels seem to target a relatively larger number of promoters than glial types (Fig 3C), but the change in expression of genes targeted by microglial TFs is more prominent than other cell types (Fig 3D). To draw a biological interpretation of the regulatory hierarchy, we performed enrichment analysis of the most confidently predicted targets (high edge-weights) of TFs using functional annotations biological process category of the human gene ontology. Interestingly, we found the top-level master regulators and the bottom-level TFs in the neuronal cell type GRNs functionally converge to regulate trans-synaptic signalling in neuronal cell types and cellular component morphogenesis in oligodendrocytes (Fig 3E). Master regulators in microglia seem to regulate small GTPase mediated signal transduction and secretion. We found that the middle-level TFs target distinct processes; synaptic signalling and neuron differentiation in excitatory neurons, secretion in

**A**

| Hierarchy | Top | | | Middle | | | Bottom | | |
|---|---|---|---|---|---|---|---|---|---|
| Cell type/Phenotype | AD | Ctrl | Random | AD | Ctrl | Random | AD | Ctrl | Random |
| Ex | 262 | 268 | 17 | 2 | 2 | 270 | 45 | 45 | 17 |
| In | 215 | 225 | 13 | 3 | 4 | 232 | 41 | 41 | 13 |
| Mic | 141 | 145 | 8 | 5 | 3 | 164 | 35 | 36 | 8 |
| Oli | 200 | 199 | 18 | 0 | 0 | 190 | 32 | 33 | 18 |

Random = 1000 trials

**Fig 3. Hierarchy analysis of cell type gene regulatory networks in AD.** (**A**) The distribution of TFs in three levels of hierarchy in AD, control and 1000 random GRNs across all four cell types. (**B**) The rewiring scores of TFs (x-axis) across all three levels of hierarchies (y-axis). The distributions of (**C**) number of promoters targeted by TFs (y-axis) and (**D**) the fold change values (log scale; y-axis) of target genes (AD versus healthy controls) of TFs at the three levels of hierarchy (x-axes). (**E**) Enrichment of gene ontology biological processes (y-axis) within targets of top, middle and bottom layers of the regulatory hierarchy across cell type networks (x-axis).

inhibitory neurons, and regulation of cell motility and cellular component movement in microglia (Fig 3E).

## Regulatory network motifs and regulatory logics

The hierarchy of our cell-type GRNs suggests that the master regulators (top TFs) regulate crucial brain-related biological functions by regulating downstream TFs, which is a highly coordinated process. For instance, various network motifs have been found in GRNs, showing such a coordination pattern in which multiple TFs co-regulate target genes. To explore the extent of coordinated TF activities in our cell type GRNs, we computed the level of over or under-representation of all possible three-node network motifs (a triplet consisting of two TFs co-regulating a target gene, previously found to be enriched in many GRNs).

As depicted in Fig 4A, our brain cell type GRNs in AD and control broadly differ in their motif composition, and AD affects some of this composition (S3 Data). For example, triplets in which two TFs target the same gene is over-represented in the microglial AD network relative to the control counterpart. On the other hand, enrichment of the motif in which two TFs are co-regulated by the same TF appears to be over-represented in microglia but under-represented in oligodendrocytes (Fig 4A), suggesting a possible disparity of TF-TF coordination across cell types. We were particularly interested in the feed-forward loops (FFL; TF1→TF2→TF3←TF1), as they have been often found to be biologically relevant in gene regulatory networks [19]. We found that FFLs are most conspicuous in the excitatory neurons and oligodendrocytes, but weakly enriched in inhibitory neurons and microglia (Fig 4A). Interestingly, a zinc-finger transcription factor specificity protein 2 (SP2) is frequently found in FFLs (Fig 4B). SP2 has been identified as a neural development gene [46], but its role in AD has not yet been elucidated. Other TFs frequently found in FFLs across most cell type GRNs include FOXP1, RFX3, ZBTB18, and PPARA (Fig 4B).

In addition to network motifs, we also investigated the cooperative logics of TFs that further reveal the TF-TF coordination mechanistically (beyond network structures like motifs). To this end, we applied our previous approach, Loregic [22], to represent gene expression relationships in FFLs using logic gate models. In particular, logic gates describe gene regulation as a two-input one-output logical process [47,48], where the expression level of regulatory factors (RFs) such as TFs are inputs and expression of target gene is the output. The logic gate has been a useful framework for studying cooperativity among RFs in human cancers, yeast and *E. coli* [19]. Thus, it would be interesting to investigate the logics behind TF cooperation in FFLs we discovered in our cell type GRNs. Fig 4C shows that AND (high target expression only when both RF1 and RF2 are high) and OR (high target expression only when either RF1 and RF2 are high) represent more than 80% of all logics across all cell types, except in microglia. Microglia has more diverse logics than other cell types, many of which involve cooperative logics (i.e., RF1 and RF2 must be particular values to activate/repress target gene). For example, compared to other cell types, a larger fraction of logics in microglia involve RF1+~RF2, which means that target expression is low only when RF1 is low and RF2 is high is (see Table S1 in [22] for explanation of these logics). An example of cooperative logics is the FFL consisting of PPARG-NFYA-CREBP which switches from uncooperative (OR) in control to cooperative (AND) in AD in the microglia network. In other words, PPARG and NFYA could be both required to active CREBP in AD, whereas either PPARG or NFYA can activate CREBP in healthy controls (Fig 4D). PPARG is a ligand-activated nuclear receptor that coordinates lipid, glucose and energy metabolism and is upregulated in AD [49]. A GWAS study suggests that NFYA gene associates with late-onset AD [50], and the CREBP gene functions in synaptic plasticity and memory formation and has been previously implicated in AD [51,52]. Thus, our

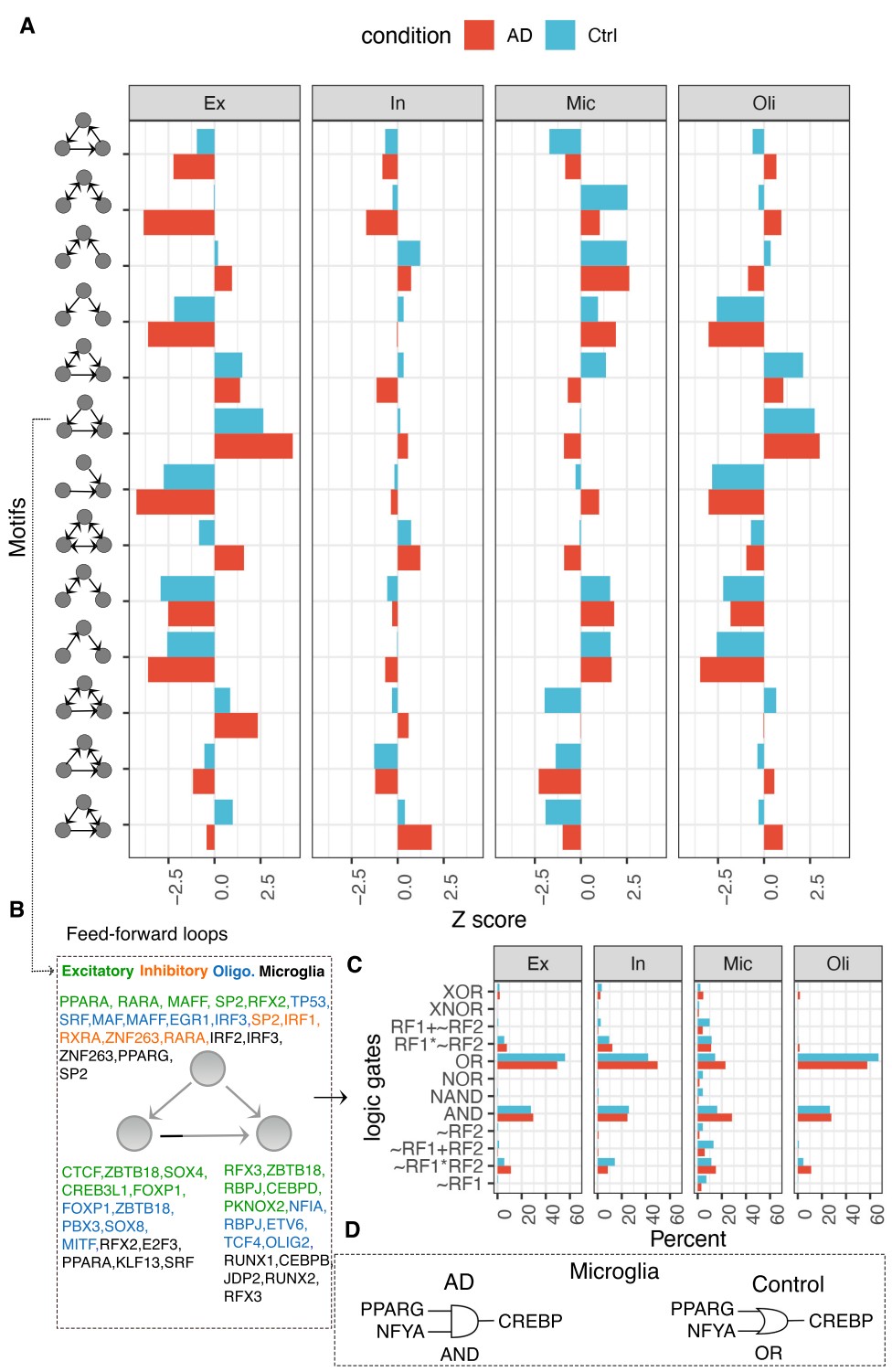

**Fig 4. Network motifs and regulatory logic across cell types in AD.** (**A**) Barplots showing the enrichment (x-axis; Z-score estimates from random networks) of various three-node triplets (y-axis) in AD (red) and control (blue) conditions across all four cell types. (**B**) Genes that frequently occur in feed-forward loops in cell type AD GRNs are depicted and colored uniquely for each cell type. (**C**) Barplot showing the frequency (x-axis) of various logic gates (y-axis) active within the feed-forward loops in AD (red) and control (blue) conditions across all four cell type networks. (**D**) Logic gate diagram showing PPARG-NFYA-CREBP triplet's AND logic in AD and OR logic in control networks of microglia.

logic analysis can further decipher the disease mechanisms of gene regulatory coordination of AD genes.

## Coregulated gene modules for AD pathology and drug repurposing

Our analysis revealed features of the regulatory hierarchy and patterns of coordinated TF action in cell type GRNs, and changes in AD. It is important to also investigate non-TF genes, as they represent the larger component of the transcriptome. These genes lie at the bottom-most layer of the regulatory hierarchy as they have no outgoing links. We reasoned that inter-rogating this highly regulated core of target genes could illuminate dysregulated AD pathways and provide a handle on network modules.

We transformed the directed GRNs into undirected networks by connecting target genes that show high levels of 'coregulation' (estimated by calculating the overlap between the predicted regulators of every pair of target genes; see Methods). Using these networks of co-regulated target genes, we tested the extent to which AD disrupts functional links between genes. We calculated the density (i.e., the ratio of observed to expected links) of the subnetworks induced by genes within carefully selected non-redundant GO BP terms. Then, comparing the densities of each GO BP term in control and AD networks allowed us to quantify the level of gain or- loss of 'cohesiveness' (i.e., interactions between GO BP genes became stronger or weaker in AD). This analysis highlighted several BP terms that significantly changed (permutation-based $P$-value < 0.001) densities across all cell types, with most in microglia (Fig 5A). For example, in microglia, interactions between genes annotated to protein-membrane transport, lipid phosphorylation, cell aging, and other sugar metabolism related terms became stronger. Whereas GO BP terms that lost cohesiveness include actin cytoskeleton organization, regulation of interleukin-2 production, and B cell proliferation, among others (Fig C in S1 File). In oligodendrocytes, interactions between genes involved in protein complex assembly, cytoskeleton organization, and neuron apoptosis became stronger, whereas interactions between genes involved in the Notch signalling pathway and transport activity became weaker (Fig D in S1 File). In inhibitory neurons, genes involved in segmentation, cell cycle, and acid transport lost cohesiveness (Fig E in S1 File). Whereas in excitatory neurons, genes involved in the cell cycle and response to biotic stimulus gained cohesiveness, while genes involved in apoptosis and response to fibroblast growth factor lost cohesiveness (Fig F in S1 File).

To explore the organization of target genes in cell type GRNs in more detail, we extracted network modules. We reasoned that the identification of modules will allow the use of mod-ules rather than individual genes as units in our investigation of novel AD risk genes. Rather than the typical approach of directly clustering gene expression data, we leveraged TF-target gene relationships embedded in our cell type GRNs to find functionally homogenous modules. Based on stringent evaluations of two clustering parameters (Fig G, Fig H, and Fig I in S1 File; see Methods), we found on average 8 modules across all cell types, and these modules are sig-nificantly enriched (1% FDR) with several GO BP terms (Fig J1 and Fig J2 in S1 File and S4 and S5 Data). We also teased out AD modules as those that were significantly enriched (1% FDR) in disease ontology terms related to AD (S6 Data). We found three AD modules each in excitatory and inhibitory neurons, and these modules are enriched in genes involved in pro-cesses related broadly to synaptic signalling, axonogenesis, and myelination (S5 Data). The two AD modules in microglia are comprised of genes involved in the regulation of GTPase activity and various immune-related processes. However, we did not detect any AD module in oligodendrocytes. Nevertheless, our analysis shows that many AD-risk genes functionally con-verge into common pathways with cell type specificity.

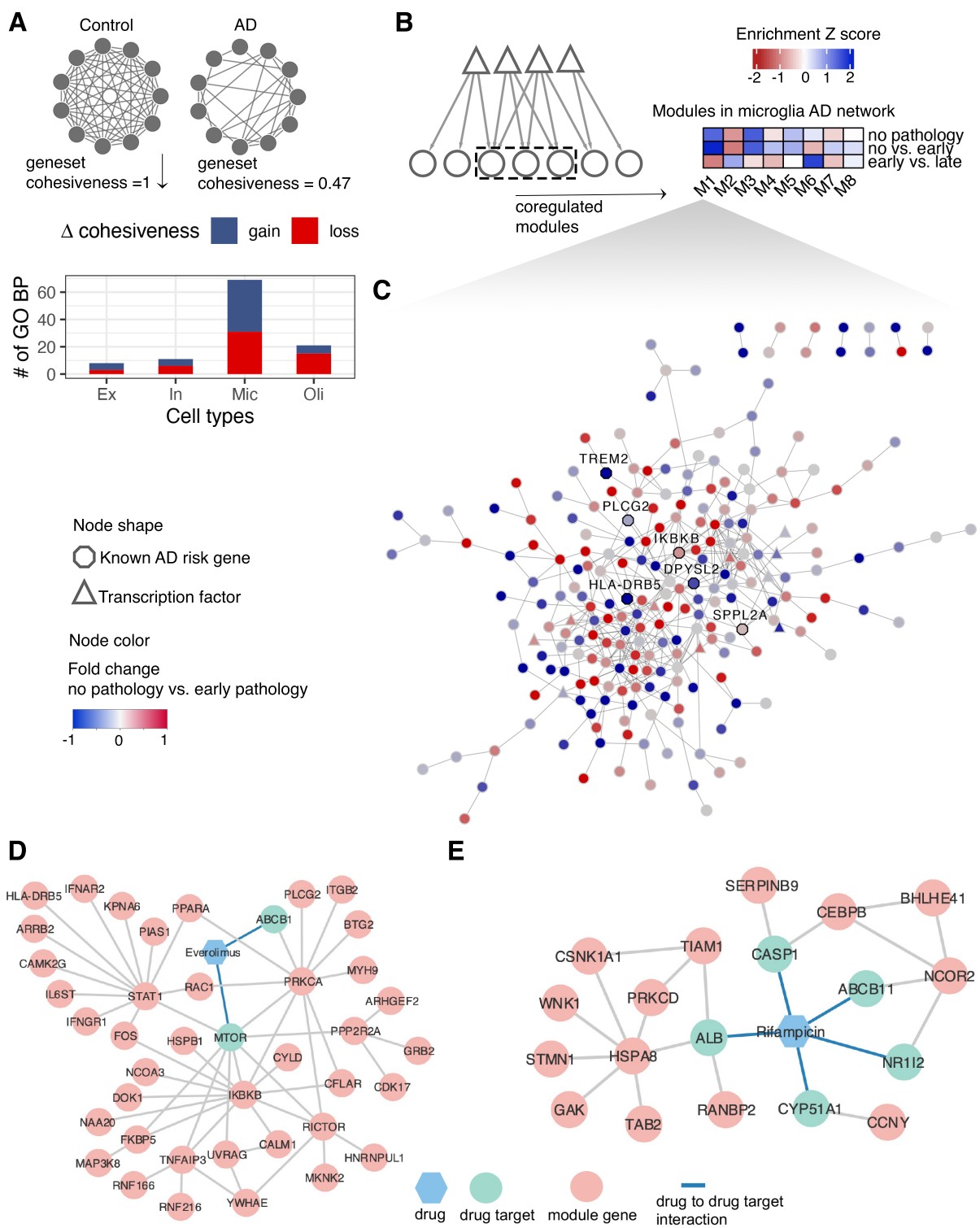

**Fig 5. Coregulated gene modules reveal cell type-specific drug-repurposed targets and gene functions in AD.** (**A**) Illustration depicting the concept of gene set cohesiveness in a network. The bar plot below shows the number of gene ontology biological process terms (y-axis) that gain (blue) or lose (red) cohesiveness between control and AD networks across all cell types (x-axis; see Methods). (**B**) A heatmap showing the enrichment of co-regulated modules of the microglia AD network within differentially expressed genes in various AD pathologies. The average fold-change of genes within each module was transformed to a Z-score to derive the enrichment score. Negative and positive Z-scores indicate down-

and up-regulation, respectively, of co-regulated modules (x-axis) in AD pathologies (y-axis). The grids of the heatmap are colored accordingly, with red indicating down-regulation and blue indicating up-regulation of the module. (**C**) Visualization of genes in module 1 of the microglia AD coregulatory network. Each circle in the plot is a gene, with TFs depicted as triangles, known AD-genes in octagons, and other genes as ellipses. Nodes are colored according to fold change values in AD pathology (early versus no pathology) as shown in the key. (**D**) Proposed mechanism-of-action for treatment of AD by everolimus using drug-target network analysis with microglia M1. (**E**) Proposed mechanism-of-action for treatment of AD by Rifampcian using drug-target network analysis with microglia M4.

We next wanted to check the response of genes within these modules in three different AD pathologies; no pathology versus pathology, no pathology versus early pathology, and early pathology versus late pathology. We focused on modules in microglia for this analysis, as it showed a more extensive change in the cohesiveness of functional gene sets (Fig 5A). We found that genes in microglia submodules 1 (M1) and 4 (M4) are more upregulated in the late stage of AD pathology compared with other microglia submodules (Fig 5B). At the same time, the disease enrichment analyses (see Methods) demonstrated that M1 and M4 are significantly associated with AD (M1with q = 3.15E-02, M4 with q = 3.59E-02). M1 is enriched with genes involved in regulation of small GTPase mediated signal transduction and immune related processes according to the GO BP annotations. Interestingly, we found TREM2 as a part of M1, including other known AD-risk genes such as PLCG2, BIN1, IKBKB, DPYSL2, SPPL2A, and HLA-DRB5 (Fig 5C). Triggering receptor expressed on myeloid cells 2 (TREM2) is a type I transmembrane protein expressed on the surface of microglia, binds to phospholipids [53] and is hypothesized to be triggering the phagocytosis of Aβ plaques [54]. A recent study showed that TREM2 deficiency results in inhibition of FAK and Rac1/Cdc42-GTPase signalling critical for microglial migration [55], testifying to the validity of M1. Neuroinflammation was proposed as one of the main mechanisms that were tightly associated with AD development [56]. KEGG pathway enrichment analysis showed that M1 was enriched with 12 immune pathways, including Fc gamma R-mediated phagocytosis, natural killer cell mediated cytotoxicity, toll-like receptor signalling pathway (Fig K1 in S1 File). Fc gamma R-mediated phagocytosis has been shown to play a role in β-amyloid dependent AD pathology [57]. Toll-like receptor 4 (TLR4) activation was previously found positively correlated with the amount of accumulated β-amyloid [58]. Furthermore, we found module M4 (Fig L in S1 File) to be enriched with genes related to immune processes, such as response to chemokines, regulation of T cell migration, cytokine regulation (Fig K2 in S1 File).

Given the valid biological link of M1 and M4 to AD pathology, we next decided to predict drug candidates based on AD-related microglia submodules M1 and M4. With the well-defined network proximity approach [59], we identified 170 and 34 candidate drugs with z_score < -2 and q < 0.05 from the total 2,891 U.S. FDA-approved or clinically investigational drugs (see Methods; S7 Data). Interestingly, one of the drugs that show significant enrichment of its targets in M1 is Donepezil (*q* value 0.008), an approved AD drug that reversibly inhibits the acetylcholinesterase enzyme. Given that the Rho GTPase activity regulates the formation of Aβ peptides during disease progression [60], our analysis raises an interesting hypothesis that the effect of Donepezil in improving the cognitive and behavioral signs and symptoms of AD might be executed via regulating GTPase signaling. Sildenafil, another top predicted drug from M1, has recently been demonstrated as one promising treatment options that showed 69% reduction in developing AD after analysing MarketScan Medicare supplemental database which included 7.23 million individuals [31]. Everolimus, an mTOR inhibitor, was another top predicted drug from M1. Everolimus was discovered to bring down both human Aβ and tau levels in the mouse model study [61]. Module M1 suggested that Everolimus's target MTOR was directly connected with multiple key AD pathology regulators, such as inhibitor of nuclear factor kappa B kinase subunit beta (IKBKB), FKBP prolyl isomerase 5 (FKBP5) (Fig

5D). One study in AD mouse model concluded that inhibiting IBKBK could help ameliorate activation of inflammatory and thus rescued cognitive dysfunction [62]. Level of FKBP5 was found to be positively correlated with AD development and FKBP5's interaction with Hsp90 accelerated tau aggregation [63]. The same study also observed that decreased amount of tau in FKBP5$^{-/-}$ mice. Rifampcian, one antibiotic drug, was top recommended according to module M4. One study found that Rifampcian was favourable for halting AD based on observations from both Aβ and tau mouse models [64]. According to our protein-protein interaction network (see Method), similarly, multiple targets of Rifampcian were the direct neighbours of multiple proteins involved in AD development (Fig 5E). CCCAT enhancer binding protein beta (CEBPB) was reported to modulate APOE's gene expression and regulated APOE4 which was one major genetic risk factor for AD in one mouse model study [65]. Protein kinase C delta (PRKCD) which was one key protein in Fc gamma receptor-mediated phagocytosis pathways was found to regulate β-amyloid dependent AD pathology [66].

## Network-based machine learning prioritizes cell-type AD-risk genes and predicts clinical phenotypes

Our analysis shows several similarities and differences in cell type GRN structures patterns across cell types and between control and AD conditions. Decomposing the GRNs into individual components using standard network analysis metrics of centrality, hierarchy and modularity outlined key genes that potentially drive changes in cell type GRNs that underpin transcriptional phenotypes of AD. However, we were still lacking a uniform scoring to rank genes according to their potential association to AD using our cell type GRNs. To facilitate this, we leveraged known AD genes in the literature and asked if the regulatory patterns that characterize these could be learned. We reasoned that our GRNs are essentially high-level features extracted by integrating single-cell multi-omics data. Thus, regulatory patterns in these GRNs can be used to train machine learning (ML) algorithms. For example, this technique of using inferred network relationships as features for a learning algorithm has helped prioritize autism and hypertension genes in humans [67,68] and stress-response TFs in plants [69].

We used the random forest algorithm to train models that learned to discriminate between known AD genes and genes unrelated to AD using their interaction patterns with TFs as features (see Methods). We wanted to compare the accuracies in predicting known AD-risk genes across cell types and between control and AD networks. The distribution of balanced accuracies in 10 independent five-fold cross-validation tests indicates that the microglia AD network most accurately predicted known AD genes compared to other networks (Fig 6A). The difference in mean accuracy between the control and AD networks of microglia is also the largest (Fig 6A). The average accuracies of these models range between 57% to 68%. The ranked lists of genes in each cell type model based on their predicted probabilities of being associated with AD can be found in S8 Data. Further, the average probability of genes that were declared as differentially expressed in control versus AD by the original authors of the dataset [2] is also relatively larger in microglial GRNs compared to other cell types (Fig M in S1 File). Genes within the top 20% of the rankings in microglia AD network are involved in immune-related processes and hemopoietic functions, cell development, and lipid metabolism (Fig 6B). These observations corroborate with previous findings. For instance, an increase of oxidative stress, changes in neuronal lipid metabolism, and synaptic dysfunction have been previously linked to early stage or overall AD pathology [70–72].

To evaluate these rankings more stringently, we asked if the expression levels of the top-ranked genes could be used to predict clinical phenotypes of AD. We utilized RNA-seq data from the ROSMAP cohort [73] to predict AD phenotypes, including Braak stages that measure

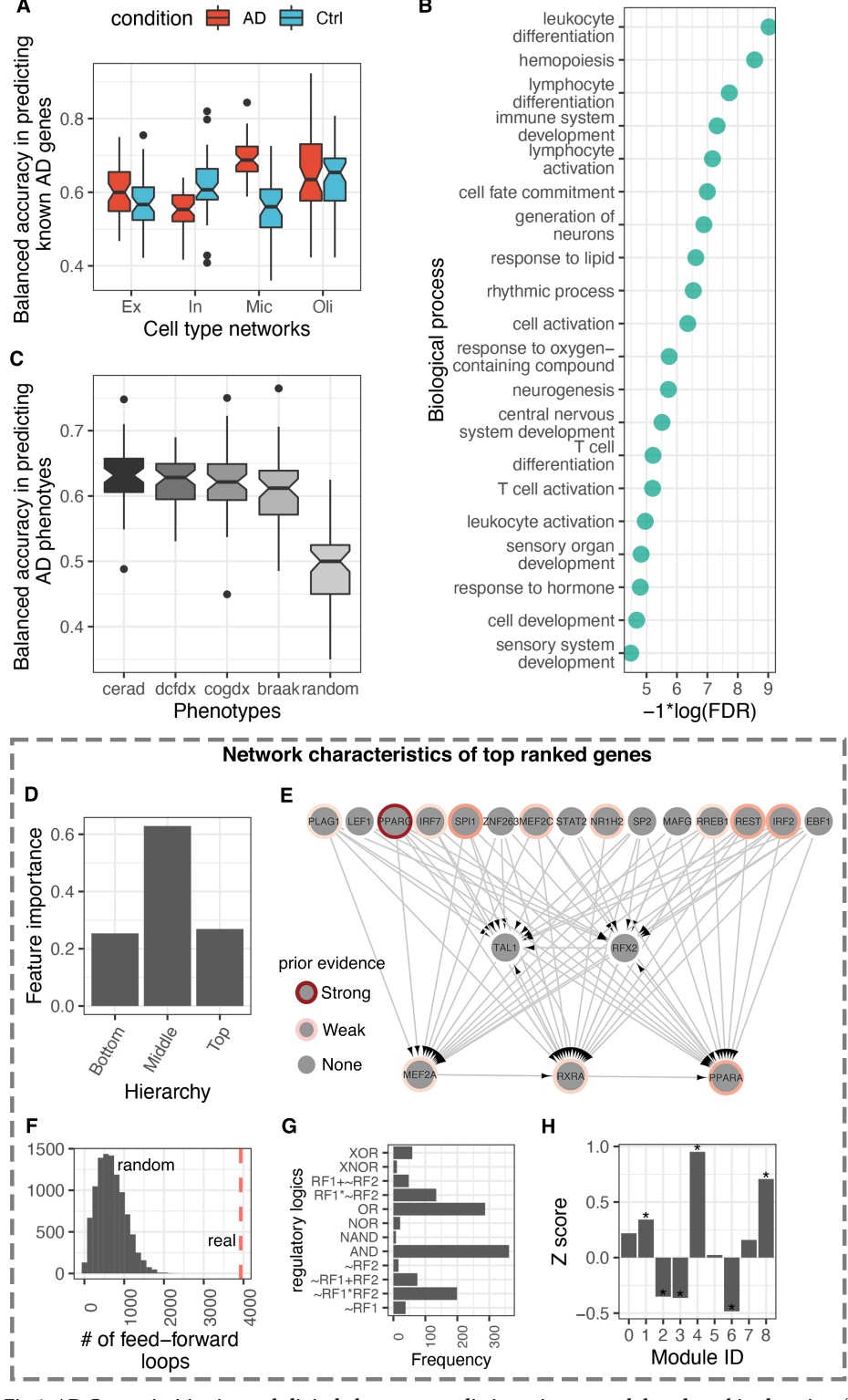

**Fig 6. AD Gene prioritization and clinical phenotype prediction using network-based machine learning. (A)** Boxplots showing the distribution of balanced accuracies (y-axis; obtained from 10 independent runs of five-fold cross-validation) in predicting known AD genes using interaction patterns in cell type GRNs as features (x-axis). **(B)** Gene ontology biological process terms enriched within the top 20% predictions in the microglia AD machine learning (ML) model. The terms are depicted along the y-axis, and the FDR corrected *p*-values are shown along the x-axis. **(C)**

Genes were sorted according to their probability of being associated with AD in the microglia ML model, and the top 5% of the sorted list was used as features to predict AD phenotypes in an independent dataset (ROSMAP). The boxplots show the distribution of balanced accuracies (y-axis) obtained from testing four AD phenotypes (see Methods) and a set of randomly selected samples (x-axis). (**D**) Average feature importance scores of TFs at the three hierarchy levels in the microglia AD network. (**E**) Visualization of the subnetwork connecting top 10% TFs with highest feature importance scores in microglia AD network. Each grey node depicts a TF with border color set along a red gradient according to the disease-gene association score given in the DisGeneNet database (based on preliminary evidence collected from independent studies). (**F**) Feed-forward loops observed within top-ranked TFs in the microglia ML model (red line) and the distribution in 1000 random networks (grey bars). (**G**) Regulatory logics observed within top-ranked TFs in the microglia. (**H**) Enrichment of top-ranked genes within coregulated genes modules in microglia (*Permutation *p* value < 0.001).

the severity of neurofibrillary tangle (NFT) pathology, CERAD scores that measure neuritic plaques, diagnosis of cognitive status (DCFDX), and cognitive status at the time of death (COGDX). Fig 6C shows the distributions of accuracy scores in predicting these phenotypes using top 5% genes in our rankings from the microglial AD model as features. Our model shows that these genes can classify AD phenotypes with more than 60% accuracy, larger than the model built using randomly selected genes (Fig 6C).

By analyzing TFs separately, we observed that those in the middle layer of the regulatory hierarchy record the highest feature importance scores (served as the best predictors in the model; Fig 6D), indicating their prominent role in regulation of gene expression in AD (S9 Data). Visualization of the subnetwork among these TFs revealed known AD genes and interaction patterns (Fig 6E). For example, we found two genes from the peroxisome proliferator-activated receptors (PPARG and PPARA) in this subnetwork. PPARs function in inflammation and immunity [74], coordinate glucose and energy metabolism [75,76], and are known to positively influence AD pathology. In addition to this, PPARA regulates genes involved in fatty acid metabolism and activates hepatic autophagy [77]. Other interesting TFs in this subnetwork include SPI1, a well-known TFs involved in microglial development and activation [78] and has been implicated in AD in GWAS [79]. Interestingly, our analysis prioritized several TFs with no previous direct associations to AD in databases. Some such examples include TAL1, RFX2, LEF1, SP2, STAT2, ZNF263, MAFG, and EBF1.

Furthermore, we found that TFs in the top of our rankings participate in a significantly larger number of feed-forward loops than expected by chance (Fig 6F). We also observed many AND and OR gates involving these TFs (Fig 6G), indicating that some of these TFs coordinate their activities to activate target gene expression. Furthermore, we found that genes at the top of our rankings are mostly overrepresented in module 4 (M4) in microglia (Fig 6H). This is interesting because M4 was also identified as disease-module with a significant number of known drug targets (Fig 5E).

## Discussion

We applied a single-cell network biology approach to compare brain cell type GRNs and examine regulatory changes that occur in AD. We obtained multi-omics data from published resources and linked TFs to TGs if the putative DNA binding motif of a given TF is located in the open and interacting promoter or enhancer region of the TG (Hi-C loops plus ATAC-seq data), and if the TF has a certain degree of coexpression with the TG in a given cell type (snRNA-seq data). We identified cell-type-specific changes in network characteristics, such as hub genes, TF hierarchies, motifs, regulatory logics, and coregulated gene modules. Further, we revealed that using those cell type networks also improved the prediction of potential novel drug targets and AD genes, which can in turn be useful in drug repurposing and predicting clinical phenotypes of AD.

Our analysis of gene centrality metrics suggests that different cell types employ a different regulatory apparatus governed by 'master regulators' that contributes to cell viability, and therefore overall brain fitness by regulating the expression of many target genes. This apparatus does not seem to be severely disrupted by the occurrence of AD, which makes sense considering previous studies on gene essentiality and lethality. Interestingly, distinct TFs with high betweenness centralities seem to modulate cell type-specific signals. Furthermore, non-TF genes, or in-degree centralities exhibit the most prominent differences between AD and control brains, indicating that these genes may contribute to pathway-level changes in AD progression. Enrichment of critical brain-specific biological processes, such as synapse organization and immune-related processes, within these genes also reflects characteristics of AD pathology. Overall, centrality analysis can delineate 'master regulators' that are potentially involved in maintaining biological processes essential for cell type function in healthy and AD individuals.

Our analysis shows that the brain GRNs are hierarchical in structure and uncovered the hierarchy height of brain TFs for the first time. Our analysis suggests that the levels on which TFs operate are generally robust to AD, and subtle changes in the expression of TFs at the top and bottom levels seem to modulate cellular signals underlying typically observed AD phenotypes. GO BP enrichment analysis also suggests that TFs in the bottom layer are involved in relevant processes like 'synapse organization', 'neuron projection development' and 'axon development' in neuronal cell types, 'neutrophil immunity', 'endocytosis', 'cell-cell adhesion' and 'actin organization' in microglia, and 'neuron projection development', 'cell morphogenesis' and 'axonogenesis' in oligodendrocytes. However, the middle-level TFs seem to be most active and perhaps cooperate and coordinate with other TFs to target a relatively larger number of genes. We noted that hierarchy height for TFs estimated using ChIP-seq datasets better reflected a tri-modal distribution [44]. In our analysis, we found fewer TFs in the middle layer and many TFs at the top layer. This could perhaps be due to many indirect TF-TG correlations that naturally arise in expression data. Another strategy to infer regulatory hierarchies more accurately would be to apply a simulated annealing procedure to the full network (including non-TF genes) to get better estimates on the actual number of hierarchies in a given network [80]. Such an analysis requires a considerable amount of computational runtime beyond our dedicated timeframe. Nevertheless, the distribution of TF hierarchies from our analysis is statistically significant compared to random networks. Moreover, whether the occurrence of fewer TFs in the middle levels is a feature of single-cell GRNs or just noise due to indirect correlations could only be evaluated based on new data from cell type-specific TF-DNA binding data.

Our TF-centric analysis indicates extensive dysregulation in the microglia network. For example, microglia has a unique set of TFs with high betweenness centrality (Fig 2B) and the largest rewiring between control and AD networks (Fig 3B). Therefore, we wanted to investigate dysregulation at the level of non-TF genes, as these genes represent the core brain pathways. Indeed, using coregulation levels as a proxy for functional relatedness, we confirmed that biological processes are most dysregulated in microglia networks (Fig 5A). This also testifies that our approach of utilizing known functional gene sets (e.g., GO terms) as biologically coherent components and using subnetwork density as a metric to gauge gene set activity is an excellent approach to highlight individual cell types. We found that the cell type networks are highly modular, and the organization of modules is largely distinct. We chose to investigate microglial module 2 further as this was the only module explicitly upregulated in the early stages of AD and statistically enriched with genes that support known AD biology. Lipid metabolism has been previously implicated in AD [70], and the role of microglia in lipid metabolism is also previously suggested [81].

Our network-proximity-based drug repurposing strategy predicted candidate AD drugs from microglial modules. We anticipate that these modules could potentially have pharmacologic applications for early intervention and drug research to target specific bio-mechanisms. Our strategy does not allow us to determine if the candidate drugs have inhibiting or activating effects on the target genes due to unsigned TF-TG edges. In future, integrating directionality such as activation or inhibition in gene regulation may help deeper understand dysregulation of gene expression in AD. Also, it is important to note that drug responses can be highly personalized. Currently, our networks are simply abstractions of high-level multi-omics data across the cohorts, and the information contained within these networks does not resolve brain-specific rewiring. Furthermore, there are many other variables that need to be accounted for (e.g., genetic susceptibility, pharmacokinetics, comorbidities, etc.) on an individual basis through extensive clinical trials. Nevertheless, our results provide a resource on cell-type regulatory networks for AD, allowing the community to further generate hypotheses and design experimental validations.

The cell type GRNs we inferred in this study, together with extensive prior genetic knowledge on AD, presented us with a unique opportunity to identify patterns of regulatory interactions that characterize AD. Inspired by previous network-based machine learning approaches, we developed an approach that leverages regulatory interactions of known AD genes as the ground truth to find other similar yet uncharacterized AD genes. Our approach correctly prioritized microglial genes related to lipid metabolism and hemopoietic function; these are well-known biological processes disrupted in AD. However, the average accuracy of our best model (~0.68) is lower than what is typically expected from such models. This lower than expected accuracy of our analysis could have arisen because we utilized network data from a single-cell type to train the models, effectively neglecting the possible functional role of other cell types in AD [82]. This could also be because our cell type networks lack chromatin interaction data in AD. Nevertheless, our gene prioritization strategy is validated using independent data from the ROSMAP study. We showed that the top 5% of genes within our ranking in the microglia network (most accurate in terms of model accuracy among all cell types) can predict clinical AD phenotypes in ROSMAP with >62% accuracy, on average of multiple five-fold cross-validation runs. We believe this is a respectable accuracy considering we made predictions on AD genes using data from a single cell type of the brain. We anticipate a further refinement of this technique in the future.

Overall, our integrated single-cell network analysis approach identified key genes and cellular themes that corroborate many aspects of AD biology. This shows that gene regulatory networks extracted from single-cell data can reveal molecular systems often hidden in gene networks derived from bulk datasets. For example, our networks revealed extensive network rewiring disrupting key biological processes mainly in microglia. As such, our approach can pinpoint cell type-specific genes that could potentially play a key role in governing disease-induced changes of pathways (e.g., lipid metabolism). As the single-cell technology further advances our ability to capture multi-modal genomic data with unprecedented precision, we anticipate that network biology applied to such single-cell functional genomics data will enhance precision medicine. Single-cell sequencing assays offer solutions to two main requisites for statistical inference of reliable gene networks; large sample size and context-specificity (unifying biological theme defined by the underlying datasets). While bulk RNA-seq datasets could provide researchers with a large enough sample size, the context-specificity is often ambiguous in publicly available datasets [83]. Single-cell technology, by design, generates volumes of data from each individual in the study. As such, pooling cell type samples from individuals is currently recommended by not required for cell type network inference. Thus, patient-specific gene networks could be possible in the coming years, which will enable us to

predict a clinical outcome better (e.g. drug response) based on network activity of target components (e.g. drug targets) [3]. Furthermore, since we already collect patient-specific data from other modalities (e.g., imaging, behavioral and clinical), fusing genetic network models with models from non-genomic modalities could resolve overlapping disease features better. Our network biology approach provides a method to investigate disease genes from single-cell data and lends itself to be used as a template for genomic feature engineering for advanced AI-based integrative models.

## Methods and materials

### Single-cell data sources and data processing

We obtained previously published single-cell gene expression data for major cell types including excitatory and inhibitory neurons, microglia, and oligodendrocyte from individuals with Alzheimer's disease pathology and healthy controls [2]. Precisely, the dataset consists of single-nucleus RNA-sequencing (snRNA-seq) of samples from the prefrontal cortex of 24 individuals diagnosed with AD pathology and 24 age-matched controls with no AD pathology. We obtained the snRNA-seq gene count matrix from synapse (Synapse: syn23446265). The original authors of the snRNA-seq dataset aggregated all 48 libraries and equalized the read depth between libraries using the CellRanger aggr pipeline before merging the data into a gene count matrix [2]. We downloaded this count matrix and further removed genes that were expressed in less than 100 cells and normalized the data using Seurat 4.0 [84]. We then applied MAGIC [85] to address dropout events by imputing the missing gene expression values and filtered lowly expressed genes to create cell type gene expression matrices. In addition, we also obtained other omics data, including cell type chromatin interaction maps (Table S5 [32]), transcription factor binding sites [34], and cell type open chromatin regions (S9 Data [33]). Note that two of these data sources (snRNA-seq and ATAC-seq) contain data for six cell types. However, the chromatin interaction data was available for only four cell types (combined for all neuronal types). Therefore, we chose to predict cell type GRNs for only the four major cell types for which all three data modalities could be obtained, as it would otherwise be difficult to tell if any observed changes between GRNs arise due to biological variation or technical variation.

### Gene regulatory network inference for brain cell types from multi-omics

We sought to integrate single-cell transcriptomic, chromatin interaction, TF binding sites, and open-chromatin regions to predict directed edges from transcription factors (TFs) to target genes (TGs). We used our scGRNom (single-cell gene regulatory network prediction from multi-omics) pipeline to perform this integration [86]. First, the scGRNom function *scGRNom_interaction* was supplied with cell type chromatin interaction data to predict all possible interactions between enhancers and promoters. Then, reference networks for each cell type were obtained by locating human TF binding sites (TFBS) within the identified interacting regions using the function *scGRNom_getTF*. Subsequently, the reference networks along with the single-cell gene expression matrix were supplied to the *scGRNom_getNt* function to predict TF-target genes for each cell type. The *scGRNom_getNt* uses elastic net regression to infer TF-target gene edges. These unsigned edges are directed from TFs to TGs and weighted according to the strength of coexpression between the given pair of TF-TG. To identify the most optimal threshold for pruning edges, we removed links with mean squared error (MSE) from elastic net regression $> 0.1$ and tested a range of absolute coefficients to further trim the edges. We found that larger values of the coefficient yield very sparse networks (very few edges and low network density), making them unsuitable for downstream analysis (Fig N in S1 File).

Therefore, we filtered the target genes with mean squared error > 0.1 and absolute coefficient < 0.01 and worked with absolute coefficient as edge weights for further analysis.

## Analysis of cell type GRN characteristics

**Centrality analysis.** Three measures of network centrality were used to gauge the importance of genes in each network. The indegree and outdegree of genes in a given network were calculated as the number of incoming TFs for each target gene and the number of target genes for each TF, respectively. The betweenness centrality was calculated by counting the number of times a given gene appears within the shortest paths of two other genes in a given network. The centrality scores for each network were scaled between 0 and 1 to make the scores comparable across cell types. To calculate fold change in centrality scores in AD versus control network of each cell type, we first replaced missing values (genes found in AD network but not in control or vice-versa) with the number that equals 1% of the smallest observed centrality score in both the networks to avoid dividing by 0. The fold change of a given gene was then calculated as the binary logarithm of the gene's normalized centrality score in the AD network divided by the control network. Genes with absolute scores > 0.5 were used for functional enrichment analysis (described below). All networks were treated as directed and the igraph R library was used to estimate gene centrality scores.

**Hierarchy analysis.** We used the hierarchy height (*h; outdegree—indegree*) of TFs to probe the direction of information flow in each network. The following analysis was performed on only TF-TF networks (TG is also a TF). The normalized *h* metric was calculated as [44]

$$h = \frac{O - I}{O + I},$$

where $O$ = outdegree and $I$ = in-degree of a TF. With this metric, TFs with *h* between 1 and 0.33 were classified as the top-level regulators, TFs with *h* between 0.33 and -0.33 were classified as the middle-level regulators, and TFs with *h* between -0.33 and -1 were classified as bottom-level regulators. The significance of the distribution of *h* metric of TFs, which was trimodal across most cell types, was calculated from the distribution of *h* in 1000 random networks (KS tests). The random networks were generated by preserving the observed edge density in each network.

**TF rewiring analysis.** To quantify the difference between sets of predicted targets of a TF in control versus AD networks, we calculated the rewiring score as [44]

$$score_{rewiring} = 1 - \frac{|Tc \bigcap Ta|}{|Tc \bigcup Ta|},$$

where |.| is the number of the set, $Tc$ and $Ta$ are the target gene sets of the control and AD networks, respectively. Thus, a high rewiring score of a TF means that its targets in the control network *(Tc)* and AD network *(Ta)* exhibit little overlap.

## Network motifs and regulatory logic analysis

Motif analysis was used to identify specific interaction patterns in the networks. We focused on subgraphs containing three genes, which were collated into 13 isomorphic classes. The number of times each class occurred in each network was recorded using the mfinder tool [87]. The Z score of the distribution was estimated from 1000 random networks. Due to the large number of networks, we set the sampling parameter to 100 to obtain a fast approximate motif analysis of the networks. To characterize TF more of action in feed-forward loops, we

applied logic circuit models using the Loregic algorithm [22]. Loregic classifies TFs into regulatory triplets (two TFs and a target gene forming an FFL in our analysis) and identifies the logic gate model (e.g. AND, OR, XOR etc.) most consistent with the cross-sample expression of each triplet. Loregic requires a binarized form of expression data as input to score the logic gate models. For each cell type gene expression matrix, we selected 100 cells with the highest variance as inputs to Loregic. 100 cells were selected to account for the uneven distribution of different cell types in the full expression matrix. This also allowed us to reduce the overall runtime of the Loregic algorithm. Loregic outputs a gate consistency score for each of the 16 possible logic gate models. For each triplet, we selected the gate model with the highest consistency score as the gate consistent for the triplet. Gates with ties in the consistency score were regarded as gate inconsistent. The statistical significance of consistent gates was estimated by replacing the target gene in each triplet with a random gene from the corresponding network and calculating the fraction of time the gate consistency score of the randomized triplet was greater than or equal to the empirical score. Consistent gates with $P < = 0.01$ were reported.

## Calculation of gene-set cohesiveness and identification of co-regulated gene modules

Our network dataset contained directed networks in which TFs are one set of nodes with outgoing links and target genes as another set of nodes with incoming links. Because TFs can also have incoming links, the networks we had at hand were essentially structured as mixed bipartite graphs. We transformed these directed graphs into undirected graphs by connecting target gene pairs if they had a considerable overlap between their predicted regulators. The overlap between the predicted regulators of a given gene pair was estimated using the Jaccard's Index (JI) and set the edge-weight. Using these weighted graphs, the gain or loss of cohesiveness within functional gene sets (GO BP terms) was estimated as follows. First, for a given gene set, a subnetwork depicting edges within the gene set was extracted. Then, the normalized network density of the subnetwork was calculated as the sum of edge weight divided by the total genes in the gene set. These operations were performed across control and AD networks of all cell types. Finally, change in gene set cohesiveness was calculated as the log ratio of density in the AD network divided by density in the control network. The statistical significance of Δ cohesiveness was calculated by randomly sampling the gene set from the background of all genes in the AD networks and calculating the picking genes from all gene sets with a fold-change greater than 0.5 were reported in Fig 5A.

Then, the adjacency matrix holding target genes in rows and columns and JI values in the cells was supplied to the WGCNA algorithm to detect coregulated gene modules [88]. The detection of reliable modules will depend on two critical parameters: the edge-weight threshold (EWT) to maintain high scoring edges and filter noise arising due to indirect regulations and the minimum module size (MMS) parameter. We wanted the MMS to be large enough (atleast 10 genes) to objectively test the functional relevance of resulting modules using statistical enrichment but not too large to include bifurcated components of large metabolic pathways into the same modules. Therefore, we tested a range of EWT (between 0.1 and 0.9) and MMS values (between 10 to 100) for every cell type network to obtain the best possible solution. We asked what combination of EWT and MMS detects the largest number of functionally relevant gene modules while retaining as many original genes as possible to avoid information loss. The functional relevance was tested by counting the fraction of detected modules that could be annotated using statistical enrichment of GO BP terms. Based on these evaluations, we found an EWT of 0.2 (20% overlap between predicted regulators of a TG-pair) and an MMS of 30 yields the best network clustering solution (Fig G, Fig H, and Fig I in S1 File). The *blockwise*

*module function* of WGCNA was invoked with 'agglomerative clustering using average linkage' as the clustering algorithm.

## Functional and disease gene enrichment analysis

The human gene ontology biological process (GO BP) annotations [89], propagated along 'is_a' and 'part_of' relationships were obtained [90]. Enrichment of query genes (e.g., top central genes, module genes etc.) within a given functional geneset (GO BP term or modules) was calculated using hypergeometric tests, using all genes present in the corresponding network as the background. The resulting p-values were corrected for multiple testing using the Benjamini–Hochberg method [91]. Note that for GO, apart from propagating parent-child relationships, we also removed geneset terms that annotate more than 500 and less than 10 genes for enrichment analysis.

## Network proximity for drug prediction

We assembled drugs from the DrugBank database relating to 2,891 compounds [92]. To predict drugs with interested modules, we adopted the closest-based network proximity measure [59] as below

$$d_{closest}(X, Y) = \frac{1}{\|X\| + \|Y\|} \left( \sum_{x \in X} \min_{y \in Y} d(x, y) + \sum_{y \in Y} \min_{x \in X} d(x, y) \right)$$

where d(x,y) is the shortest path length between gene x and y from gene sets X and Y, respectively. In our work, X denotes the interested modules, Y denotes the drug targets (gene set) for each compound. To evaluate whether such proximity was significant, the computed network proximity is transferred into z score form as shown below:

$$Z_{d_{closest}} = \frac{d_{closest} - \mu_d}{\sigma_d}$$

Here, $\mu_d$ and $\sigma_d$ are the mean and standard deviation of permutation test with 1,000 random experiments. In each random experiment, two random subnetworks $X_r$ and $Y_r$ are constructed with the same numbers of nodes and degree distribution as the given 2 subnetworks X and Y separately, in the protein-protein interaction network.

## Protein-protein interactome (PPI) network

To build the comprehensive human interactome from the most contemporary data available, we assembled 18 commonly used PPI databases with experimental evidence and the in-house systematic human PPI that we have previously utilized: (i) binary PPIs tested by high-throughput yeast-two-hybrid (Y2H) system [93]; (ii) kinase-substrate interactions by literature-derived low-throughput and high-throughput experiments from KinomeNetworkX [94], Human Protein Resource Database (HPRD) [95], PhosphoNetworks [96], PhosphositePlus [97], DbPTM 3.0 and Phospho.ELM [98]; (iii) signaling networks by literature-derived low-throughput experiments from the SignaLink2.0 [99]; (iv) binary PPIs from three-dimensional protein structures from Instruct [100]; (v) protein complexes data (~56,000 candidate interactions) identified by a robust affinity purification-mass spectrometry collected from BioPlex V2.0 [101]; and (vi) carefully literature-curated PPIs identified by affinity purification followed by mass spectrometry from BioGRID [102], PINA [103], HPRD [104], MINT [105], IntAct [106], and InnateDB [107]. Herein, the human interactome constructed in this way includes 351,444 PPIs connecting 17,706 unique human proteins.

## Machine-learning model for AD-gene prioritization

We sought to utilize network connectivity patterns in cell type regulatory networks to make predictions on disease-gene associations. First, we downloaded known disease-gene associations listed in the DisGenNet database [108] and extracted all genes linked with the keyword 'Alzheimer'. The DisGenNet database ranks gene-disease associations using a metric that quantifies the level of evidence in published literature. 16% (3481 out of 21666) of all genes in the database are linked with AD, with gene-disease associations scores ranging from 0.01 (not strong evidence) to 0.9 (strong evidence). We selected AD genes with scores greater than 0.1 (top 20%) as positive examples to build the binary classifiers. Then, rather than randomly selecting negative samples, we further analyzed the DisGenNet database to identify genes that are likely not associated with AD. To do this, we calculated overlaps between diseases and selected genes strongly associated with diseases that have minimal overlaps with AD (disease-disease Jaccard's overlap < 0.1). From this pool of 'likely not AD-associated' genes, we randomly selected negative examples equal to the number of positive examples to build classifiers not biased by class-size. Then, each GRN was transformed into a non-symmetrical adjacency matrix *A*, with TFs (*i*) in columns and TGs (*j*) in rows and the cell *Aij* containing the predicted edge score (absolute coefficient of elastic net regression from scGRNom) of the corresponding TF-TG pair. The subset of *A* with rows containing our positive and negative samples was extracted as the feature matrix, *F*. To include TFs that do not have any in-degrees (not regulated by other TFs in our networks) in *F*, we assigned an edge score equal to 1% of the minimum edge score in the corresponding network. This allowed us to label TFs with no upstream regulators and include them in prediction models. Then, using the vector of edge scores of each sample in *F* as the feature vector, we trained a random forest classifier to discriminate between positive and negative samples. The balanced accuracy of the model was tested using 10 independent runs of five-fold cross-validation. The average balanced accuracy (total 50 trials) was recorded and plotted. The predicted probability of class output from the classifier was used to rank all genes. The feature importance score was measured as the Gini impurity. The Gini impurity metric estimates the probability of classifying a sample incorrectly, and is calculated as

$$G = \sum_{i=1}^{C} p(i)(1 - p(i)),$$

where *C* is the total number of classes (2 in our case) and *p(i)* is the probability of picking a sample in class *i*. The accuracy and *G* were recorded for each cell type GRN in both conditions. The most accurate cell type model was chosen as the one with the highest average accuracy (AD microglia network in our study) and used to predict the probability of AD association of the remaining unlabelled genes along with TFs with the largest feature importance scores.

## Prediction of clinical phenotypes

To predict AD phenotypes, we utilized the original RNA-seq data from the ROSMAP study (55,889 Ensembl gene ids for 640 post-mortem human samples) on an Alzheimer's disease case-control cohort for the Dorsolateral Prefrontal Cortex (DLPFC) brain region. We obtained permission from ROSMAP to use this data (available on synapse.org (ID: syn3219045). We mapped the Ensembl genes ids to Entrez gene identifiers, averaged the gene expression values for Ensembl gene identifiers that mapped to the same Entrez identifiers, and removed unmapped Ensembl identifiers. Ultimately, we found 26,017 genes (with unique Entrez IDs). Only 638 out of 640 individual RNA-Seq samples mapped to population phenotypes. Our final

DLPFC dataset thus contained gene expression values for 26,017 genes for 638 samples. Then, using normalized gene expression values of top 5% ranked genes from the microglia AD-gene classification model (described above) as the feature vectors, we trained random forest classifiers to predict various AD phenotypes. The following coding was used: cogdx (4 and 5 versus 1), braak (0,1,2 versus 5,6), and cerad (1 versus 3,4). The classifier accuracy was evaluated using 10 independent runs of five-fold cross-validations, as described above.

## Supporting information

**S1 Data. Gene centralities across all cell type gene regulatory networks (GRNs).**
(XLSX)

**S2 Data. Hierarchy of TFs across all cell type GRNs.**
(XLSX)

**S3 Data. Results of motif enrichment of cell type GRNs.**
(XLSX)

**S4 Data. Gene module memberships.**
(XLSX)

**S5 Data. GO enrichment results of network modules.**
(XLSX)

**S6 Data. Disease ontology enrichment of network modules.**
(XLSX)

**S7 Data. Drug prediction results.**
(XLSX)

**S8 Data. Predicted probabilities of network-based AD gene classifiers.**
(XLSX)

**S9 Data. Feature importance scores of TFs in microglia AD network.**
(XLSX)

**S1 File. Figs A-N. Fig A**. (**1**) Distribution of different edge types within cell type GRNs. (**2**) An upset plot showing overlaps between the top 10% genes with largest in-degree. The filled dots in the centre matrix indicate the comparison between the respective sets (along the x-axis), and the bars on the top show size of the intersection. Blue and red rows indicate control and AD, respectively. The red arrow at the bottom shows overlaps between both neuronal types across the two phenotypes. (**3**) Overlaps between central genes in two independent datasets (Mathys et al. and Lake et al.). (**4**) Heatmaps depicting correlation in gene centrality scores between the full dataset and a reduced dataset consisting 50% of the original samples in the snRNAseq dataset. **Fig B**. **Hierarchy analysis of cell type GRNs.** (**1**) Sankey plots showing overlaps between regulatory hierarchies of cell type control and AD GRNs (Blue: top-level, green: middlelevel, red: bottom-level). Overlaps between (**2**) top-level, and (**3**) middle-level TFs across cell types. **Fig C**. Barplot showing GO BP terms that gain or loss cohesiveness measured as change in network density between control and AD networks of microglia. **Fig D**. Barplot showing GO BP terms that gain or loss cohesiveness measured as change in network density between control and AD networks of oligodendrocytes. **Fig E**. Barplot showing GO BP terms that gain or loss cohesiveness measured as change in network density between control and AD networks of inhibitory neurons. **Fig F**. Barplot showing GO BP terms that gain or loss cohesiveness measured as change in network density between control and AD networks of

excitatory neurons. **Fig G**. Barplots showing the number of co-regulated gene modules (left y-axis) detected at various levels of edge-weight threshold (right y-axis) and minimum module size cut-off (top x-axis) across cell types (x-axis). **Fig H**. Barplots showing the number of genes included in modules (left y-axis) detected at various levels of edge-weight threshold (right y-axis) and minimum module size cut-off (top x-axis) across cell types (x-axis). **Fig I**. Barplots showing the fraction of functionally annotated modules (left y-axis) detected at various levels of edge-weight threshold (right y-axis) and minimum module size cut-off (top x-axis) across cell types (x-axis). **Fig J**. Barplots showing (**1**) the number of modules detected in cell type AD and control networks, and (**2**) the number of enriched GO BP within those modules. **Fig K**. (**1**) KEGG pathway enrichment of module M1 and (**2**) GO BP enrichment analysis of module M4 in microglia. Pathways/processes are shown on the y-axis and the adjusted pvalue of enrichment is shown along the x-axis. **Fig L**. Genes in module M4 of microglia AD network. **Fig M**. Predicted probabilities of differentially expressed genes (DEG) being associated with AD based on the random forest-based classifier. **Fig N**. Distribution of (**1**) edges, (**2**) TFs and (**3**) TGs within various threshold of absolute coefficients from the elastic net regression. (PDF)

## Acknowledgments

The authors wish to thank all members of the Wang lab for insightful discussions on the topic.

## Author Contributions

**Conceptualization:** Daifeng Wang.

**Data curation:** Chirag Gupta, Daifeng Wang.

**Formal analysis:** Chirag Gupta, Jielin Xu, Ting Jin, Saniya Khullar, Xiaoyu Liu, Sayali Alatkar.

**Funding acquisition:** Daifeng Wang.

**Investigation:** Chirag Gupta, Feixiong Cheng, Daifeng Wang.

**Methodology:** Chirag Gupta, Jielin Xu, Feixiong Cheng, Daifeng Wang.

**Project administration:** Daifeng Wang.

**Resources:** Chirag Gupta, Jielin Xu, Feixiong Cheng, Daifeng Wang.

**Software:** Chirag Gupta, Jielin Xu, Ting Jin, Saniya Khullar.

**Supervision:** Feixiong Cheng, Daifeng Wang.

**Validation:** Chirag Gupta.

**Visualization:** Chirag Gupta, Jielin Xu, Saniya Khullar.

**Writing – original draft:** Chirag Gupta, Jielin Xu, Saniya Khullar, Feixiong Cheng, Daifeng Wang.

**Writing – review & editing:** Chirag Gupta, Jielin Xu, Saniya Khullar, Feixiong Cheng, Daifeng Wang.

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
