## [Decision Letter · Decision Letter 0]

1 Mar 2022

Dear Dr Wang,

Thank you very much for submitting your manuscript "Single-cell network biology characterizes cell-type gene regulation for drug repurposing and phenotype prediction in Alzheimer's disease" for consideration at PLOS Computational Biology.

As with all papers reviewed by the journal, your manuscript was reviewed by members of the editorial board and by several independent reviewers. In light of the reviews (below this email), we would like to invite the resubmission of a significantly-revised version that thoroughly addresses all reviewers' concerns.

We cannot make any decision about publication until we have seen the revised manuscript and your response to the reviewers' comments. Your revised manuscript is also likely to be sent to reviewers for further evaluation.

Sincerely,

Qing Nie

Associate Editor

PLOS Computational Biology

Jian Ma

Deputy Editor

PLOS Computational Biology

Reviewer's Responses to Questions

**Comments to the Authors:**

**Reviewer #1: **The study by Gupta and colleagues carried out multiple levels of advanced network analyses to characterize cell-type level gene regulatory networks in Alzheimer’s brains and controls and to identify the changes. The authors also developed new machine learning models to classify and prioritize Alzheimer’s risk genes, including finding new candidates, and studied how the prioritized genes help predict clinical phenotypes. They further applied their recent network drug repurposing approaches to the regulatory networks to identify potential drugs. The study is highly original, comprehensive, novel, and represents significant computational advance. The network methods are very interesting to computational investigators and the findings add significant values for understanding the gene regulatory network changes and gene-phenotype relationship in Alzheimer’s brains. The manuscript is written very well and easy to follow. It is relatively long because it covers several areas. Below are some technical concerns and suggestions for improving the work.

Major concerns:

1. The biological relationship of the source data needs to be described. The scRNA-seq data for controls and Alzheimer’s diseases contained additional cell types not used in this study. Why were they excluded? Are the Hi-C data (in ref #32) for the same cell types defined as in the snRNA-seq data? More generally, how cell types from one data source compared to the cell types in another? Will that “mismatch” have any potential influence on the results? Ref #85 also has Hi-C data, any reason Hi-C data from one study is better than the other?

2. The stability and robustness of the results need to be addressed. In the study, snRNA-seq data from all 24 patients and 24 controls were combined, but there must be some variations among the brains. Can the authors check how their networks vary when a subset of the samples is used? In related, maybe the authors can perform an analysis with 12 patients x 12 patients, to show what null result of “changing” gene regulatory networks looks like.

3. The choice of parameters or thresholds needs to be clarified. For example, in the scGRN analysis, “we filtered the target genes with mean square error > 0.1 and absolute elastic net coefficient < 0.01.” For people not familiar with this, it is not clear why 0.1 and 0.01 were used.

Minor comments:

4. Do the scGRN contains both activating and inhibiting TF-target relationships?

5. Figure 1D is missing.

6. The description of Figure S1A (pg 5) seems inconsistent with what is in the figure.

7. Figure 2 legends for 2E and 2F need to be swapped.

8. Figure 3, can the authors make better visualization to support “tri-model” in 3A? The finding of so many master regulators is counterintuitive. 3D and 3E show average changes. Do they come with some variations?

9. In the drug repurposing analysis, do the connections of a drug to genes consider the directions of effects, i.e., activating or inhibiting genes?

10. In pg 21, which supplementary figures were meant to be cited in “Sup Fig.?”

11. In the scNET GitHub page, the authors list dependent libraries. Can they include version requirement?

**Reviewer #2: **Authors in this manuscript present an interesting network approach to characterize cell-type gene regulation for drug repurposing

and phenotype prediction in Alzheimer's disease at a single-cell resolution. While it is interesting and timely, I have the following major concerns.

I am very confused about the central idea of this manuscript. Is this a new computational methodology or a biological analysis paper? There is no benchmarking and software releasing if it is a new computational method. If this is an analysis paper, then there is no validation of the results and reasoning of methodology selections.

I have some major concerns regarding the network construction part. There is no intuition and details about how the network is constructed. How different multi-omics data are used? What are the basic summery of chromatin interactions? How did the authors process the data (if authors downloaded it as public data, where are they and what is the pre-processing)? Cell-type resolution of the network analysis is the paper's basis, but there is no QC of the network at all.

The authors constructed separate networks on AD and control patients. However, how did the scRNA-seq data are normalized across different individuals? There is no description of the scRNA-seq data itself. To what degree of the network differences might be due to biases?

Some of the biological processes in Fig6 is very different to interpret as AD-related. Is there any validation of these prioritizations?

Drug responses are highly personalized. How much variation is expected at the network level in AD patients?

some of the analysis is difficult to be linked to the goal of this manuscript - drug repurposing and phenotype prediction in AD. For instance, how did the network motifs contribute to this goal?

**Have the authors made all data and (if applicable) computational code underlying the findings in their manuscript fully available?**

Reviewer #1: Yes

Reviewer #2: **No: **It is very confusing whether this is a method paper or analysis paper

PLOS authors have the option to publish the peer review history of their article (what does this mean?). If published, this will include your full peer review and any attached files.

Reviewer #1: No

Reviewer #2: No
---

## [Decision Letter · Decision Letter 1]

7 Jun 2022

Dear Dr Wang,

We are pleased to inform you that your manuscript 'Single-cell network biology characterizes cell type gene regulation for drug repurposing and phenotype prediction in Alzheimer’s disease' has been provisionally accepted for publication in PLOS Computational Biology.

Best regards,

Qing Nie

Associate Editor

PLOS Computational Biology

Jian Ma

Deputy Editor

PLOS Computational Biology

Reviewer's Responses to Questions

**Comments to the Authors:**

Reviewer #1: My previous comments were sufficiently addressed.

Reviewer #2: All my questions have been addressed

**Have the authors made all data and (if applicable) computational code underlying the findings in their manuscript fully available?**

Reviewer #1: Yes

Reviewer #2: None

PLOS authors have the option to publish the peer review history of their article (what does this mean?). If published, this will include your full peer review and any attached files.

Reviewer #1: No

Reviewer #2: No

---

## [Editor Report · Acceptance letter]

12 Jul 2022

PCOMPBIOL-D-22-00116R1 

Single-cell network biology characterizes cell type gene regulation for drug repurposing and phenotype prediction in Alzheimer’s disease

Dear Dr Wang,

I am pleased to inform you that your manuscript has been formally accepted for publication in PLOS Computational Biology. Your manuscript is now with our production department and you will be notified of the publication date in due course.

With kind regards,

Zsofi Zombor
